# Antimicrobial Peptides (AMP) in the Cell-Free Culture Media of *Xenorhabdus budapestensis* and *X. szentirmaii* Exert Anti-Protist Activity against Eukaryotic Vertebrate Pathogens including *Histomonas meleagridis* and *Leishmania donovani* Species

**DOI:** 10.3390/antibiotics12091462

**Published:** 2023-09-19

**Authors:** András Fodor, Claudia Hess, Petra Ganas, Zsófia Boros, János Kiss, László Makrai, Károly Dublecz, László Pál, László Fodor, Anna Sebestyén, Michael G. Klein, Eustachio Tarasco, Manjusha M. Kulkarni, Bradford S. McGwire, Tibor Vellai, Michael Hess

**Affiliations:** 1Department of Genetics, Institute of Biology, Eötvös Loránd University, Pázmány Péter. sétány 1C, H-1117 Budapest, Hungary; boroszsofia@student.elte.hu (Z.B.); vellai.tibor@ttk.elte.hu (T.V.); 2Clinic for Poultry and Fish Medicine, Department for Farm Animals and Veterinary Public Health, University of Veterinary Medicine (Vetmeduni Vienna), 1210 Vienna, Austria; claudia.hess@vetmeduni.ac.at (C.H.); petra.ganas@bfr.bund.de (P.G.); 3Agribiotechnology and Precision Breeding for Food Security National Laboratory, Department of Microbiology and Applied Biotechnology, Institute of Genetics and Biotechnology, Hungarian University of Agriculture and Life Sciences, Páter Károly utca 1, H-2100 Gödöllő, Hungary; kiss.janos@uni-mate.hu; 4Autovakcina Kft., H-1171 Budapest, Hungary; makrai.laszlo@univet.hu; 5Institute of Physiology and Nutrition, Georgikon Campus, Hungarian University of Agriculture and Life Sciences (MATE), Deák Ferenc utca 16, H-8360 Keszthely, Hungary; dublecz.karoly@uni-mate.hu (K.D.); pal.laszlo@uni-mate.hu (L.P.); 6Department of Microbiology and Infectious Diseases, University of Veterinary Medicine, H-1143 Budapest, Hungary; fodor.laszlo@univet.hu; 7First Department of Pathology and Experimental Cancer Research, Semmelweis University, H-1085 Budapest, Hungary; sebestyen.anna@semmelweis.hu; 8USDA-ARS & Department of Entomology, The Ohio State University, 13416 Claremont Ave, Cleveland, OH 44130, USA; klein.10@osu.edu; 9Department of Soil, Plant and Food Sciences, University of Bari “Aldo Moro”, Via Amendola 165/A, 70126 Bari, Italy; eustachio.tarasco@uniba.it; 10Division of Infectious Diseases, Department of Internal Medicine, The Ohio State University, Columbus, OH 43210, USA; manjumkulkarni@gmail.com (M.M.K.); brad.mcgwire@osumc.edu (B.S.M.)

**Keywords:** non-ribosomal anti-microbial peptides (NR-AMP), entomopathogenic nematode-symbiont bacteria (EPB), *Xenorhabdus*, *X. budapestensis*, *X. szentirmaii*, *X. innexii*, *Photorhabdus* (*P. luminescens TT01*), CFCM (cell-free conditioned culture media), kinetoplastid protozoa parasite, *Leishmania*, extracellular avian pathogens, *Histomonas meleagridis*, probiotic potential

## Abstract

Anti-microbial peptides provide a powerful toolkit for combating multidrug resistance. Combating eukaryotic pathogens is complicated because the intracellular drug targets in the eukaryotic pathogen are frequently homologs of cellular structures of vital importance in the host organism. The entomopathogenic bacteria (EPB), symbionts of entomopathogenic–nematode species, release a series of non-ribosomal templated anti-microbial peptides. Some may be potential drug candidates. The ability of an entomopathogenic–nematode/entomopathogenic bacterium symbiotic complex to survive in a given polyxenic milieu is a coevolutionary product. This explains that those gene complexes that are responsible for the biosynthesis of different non-ribosomal templated anti-microbial protective peptides (including those that are potently capable of inactivating the protist mammalian pathogen *Leishmania donovanii* and the gallinaceous bird pathogen *Histomonas meleagridis*) are co-regulated. Our approach is based on comparative anti-microbial bioassays of the culture media of the wild-type and regulatory mutant strains. We concluded that *Xenorhabdus budapestensis* and *X. szentirmaii* are excellent sources of non-ribosomal templated anti-microbial peptides that are efficient antagonists of the mentioned pathogens. Data on selective cytotoxicity of different cell-free culture media encourage us to forecast that the recently discovered “easy-PACId” research strategy is suitable for constructing entomopathogenic-bacterium (EPB) strains producing and releasing single, harmless, non-ribosomal templated anti-microbial peptides with considerable drug, (probiotic)-candidate potential.

## 1. Introduction

Anti-microbial peptides (AMPs) are any polyamide, or even their bio-co-polymer with esters, thioesters, or otherwise modified backbone, that can be made on a contemporary chemical peptide synthesizer [1]. This provides an option of taking advantage of using the remarkable toolkit of quantitative structure–activity relation (QSAR) [2,3] in designing novel anti-microbial-active sister molecules of any newly discovered natural drug-candidate AMP molecule. As for their biosynthesis, peptides of anti-microbial potential can be either ribosomal encoded (RP) single gene products [4] or enzymatically biosynthesized non-ribosomal templated peptides (NRP) [5], (NR-AMPs) [6]. Considering that there are already >100 peptide-based drugs that have been of clinical use (see [7]), an lookout at that field may be advisable. As for history, we refer to the review of Zasloff [8], who outlined those discoveries that can be considered as milestones, like cecropins [9], defensins [10,11], maganins [12], and add the proline-rich AMP (PrAMP family) [13,14] to the still incomplete list. Many derivatives of discovered natural AMPs have been chemically synthesized and or modified, and there are several excellent reviews [15,16,17,18,19] beyond the scope of this paper.

The two selected targets here are *Leishmania* [20], a mammalian pathogen, transmittable by insects like cave-dwelling sand flies [21], and *Histomonas meleagridis*, the etiological agent of the life-threatening histomonosis (syn. blackhead disease) of poultry species such as turkeys, chickens, and pheasants, first described in 1893 [22,23], and have been studied in detail until the mid of last century [24]. 

In the actual study we aimed to investigate if any of those biosynthetic NR-AMPs, like fabclavine [25,26], phenazine [27], and others, produced by entomopathogenic–nematode symbiont bacteria (EPB) we have been studying [28,29] were suitable as anti-protozoal drug candidates against the intracellular parasite *Leishmania* or the extracellular parasite *H. meleagridis*. The arguments for choosing in vitro EPB liquid cultures as appropriate sources of efficient AMPs for this purpose are that obligate bacterial symbionts of entomopathogenic–nematode species synthesize and release several non-ribosomal hybrid peptides [30] with large target specificities. These mainly serve to provide well-balanced pathobiome conditions for this symbiosis in polyxenic soil and cadaver environments [31]. These peptides are also considered potential sources of potent natural anti-microbial compounds [30]. The motivation for choosing in vitro EPB liquid cultures as a source of efficient AMPs for this purpose is the proven activity on Gram-positive [32,33] and Gram-negative [34] pathogenic bacteria, oomycetes, and different plant pathogens [35,36] in our previous studies. Since we have reasons to believe that some of the autoclaved *Xenorhabdus* cultures may not be harmful or toxic when added as a food supplement, while some anti-microbial ingredients retain active [6], we consider the potential for probiotic applications, at least on an experimental level. 

The eukaryotic pathogens studied here include the *Leishmania donovanii* species, kinetoplastid protozoa [37], and the causative agent of human and canine leishmaniasis [38,39,40].

*Leishmania* are intracellular parasites that target professional phagocytes (macrophages and dendritic cells) [41] and human infection leads to diverse forms of disease from singular or diffuse cutaneous lesions (cutaneous leishmaniasis), invasive destruction of mucous membranes (mucocutaneous leishmaniasis), or dissemination to the liver, spleen, and bone marrow (visceral leishmaniasis) [42,43]. While mainly relegated to endemic foci in poor rural areas in the third world, the infections have a global impact due to human migration, climate change, and anthropogenic disturbance, causing significant worldwide health and economic burdens [44]. There are currently no approved vaccines available [45], leaving control of leishmaniasis reliant on chemotherapy [46]. The mainstays of therapy are antimonial derivatives [47] of amphotericin B (AmB), a polyene macrolide antibiotic derived from actinomycetes [48], which can be renally toxic and are not widely available in poor rural areas. Increasingly, there are issues with leishmanial infections that are resistant or refractory to antimony and amphotericin therapy. Miltefosine is the first oral agent approved for the treatment of leishmaniasis; however, there have been descriptions of miltefosine resistance [46]. There is an obvious need for the development of newer, less toxic agents to combat these infections.

The other chosen pathogen, *H. meleagridis*, belongs to the Dientamoebidae, order *Tritrichomanidida*. Infections in turkeys may cause nearly 100% mortality whereas outbreaks in chickens, peafowl, quail, and pheasants are more often marked by morbidity and subsequent recovery. *H. meleagridgridis* is carried by the eggs of the cecal worm *Heterakis gallinarum*, enabling them to survive for long periods in the soil as a source of infection [22,23,24]. In the EU and USA, there are currently no drugs available for the treatment of blackhead disease [22,24]. (As for molecular taxonomy and identification, the 5.8S, ITS-1, and ITS-2 rRNA regions were first sequenced [49]. Since then, the complete, annotated *Histomonas* reference genome sequence became available [50]).

Since the introduction of effective drugs into the market in the middle of the previous century, there was no practical need for further research. Following the recent ban of available drugs, research programs with new profiles were set up in various places focusing on different features of the parasite and the disease. 

Consequently, the poultry industry works without approved prophylactics, therapeutics, or vaccines to combat histomonosis [51].

Recently, numerous chemical and botanical compounds were also tested for their efficacy against *H. meleagridis*, with varying outcomes. One of the explanations for these half-failures/half-successes may be the complicated in vitro *Histomonas* culturing technique. In this context, it is of high importance that *H. meleagridis* relies on live bacteria and *Escherichia coli* strongly supports the growth of *H. meleagridis* in a monoxenic culture without influencing its pathogenicity [52]. In the previously mentioned study, a clonal cultures of *H. meleagridis* was further optimized to obtain a monoxenic culture in a liquid medium [52]. For this, the fecal flora was exchanged for defined bacterial strains by selective destruction of the initial bacteria with various antibiotics, keeping the flagellate alive. *E. coli* was found to strongly support the growth of the parasite, whereas *Salmonella enterica* serovar Typhimurium and *Pseudomonas aeruginosa* were less efficient. Whether the special feature of the parasite’s intricate interplay with bacteria in vitro and in vivo can be considered mutualistic or a predator-prey one is an open question [50,51,52,53,54,55,56,57,58,59,60]. (We are shared about it). The option of the monoxenic culturing technique may open a door to preliminary attempts to protect birds with cultured attenuated histomonads, forecasting the possibility of vaccination [56].

After banning all previously used prophylactic and therapeutic drugs against *H. meleagridis* in the USA and the European Union, benzimidazoles were tested but found inactive [61]. Similar half-successes were reported about various herbal substances, some of which showed good efficacy in vitro but failed in vivo [62]. Reduced sensitivity of *H. meleagridis* to nitarsone in vitro and in vivo was also shown [63]. 

A publication talked about that, identical monoxenic settings for cultures of the same *H. meleagridis* clonal strain, in its virulent low-passage and attenuated high-passage form, enabled a comparative analysis of parasite characteristics [64], but the conclusions severely contradict a previously cited, unambiguously reliable publication [59]. This debate is out of scope of the present paper.

Herein, we compared the spectrum of in vitro anti-microbial/-prozozoal activity of cell-free conditioned culture media (CFCM) from three EPB species, together with testing against *Leishmania donovanii*, (together with *Trypanosoma cruzi*, as an expectedly negative control), *Histomonas meleagridis*, and also against a panel of clinical bacterial and fungal isolates, as positive controls. Considering the roles of the bacterial symbiont in the EPN/EPB symbiosis, we suppose that ani-protozoal compounds must coexist with other anti-microbial compounds in the CFCM of EPBs. Therefore, for reliable bioassays, we should use appropriate controls. Two main aspects should be taken into consideration: (A) To be able to distinguish between the specified antiprotist activity and general cytotoxicity of the future drug-candidate anti-microbial compounds in the CFCM. (B) To be able to identify the special antiprotist compounds. In this paper, we describe the very first steps toward these goals. For (A), one has to consider that the lack of uniformity regarding the choice of cell types for cytotoxicity assays may lead to incomparable and inconclusive data. In vitro assays relying solely on non-phagocytic cell models may not represent a realistic result, as the effect of an anti-leishmanial agent should ideally be presented based on its cytotoxicity profile against reticuloendothelial system cells [65]. In the *Leishmania* studies, we used the macrophage cell line J774A.1 [66,67,68], as a cytopathogenicity control test organism (see Section 2 Materials and Methods). In the *Histomonas* studies, we used the permanent chicken liver cells (LMH leghorn male hepatoma) cell line [6,69,70] as a cytopathogenicity control test organism (see Section 2 Materials and Methods). For (B), one has to take into consideration the following points: each NR-AMP-producing EPB cell is in the so-called primary (1^0^) phase, or Phase 1 [71]. In other terms, the phenotypic precondition for antibiotic production (as well as the capability of symbiosis) of the EPB species is the primary phase [30,72,73]. For NR-AMP production, it means that in a liquid culture of the primary (1^0^) cells (of each of the studied EPB species, strain, and isolate), more than one biosynthetic AMP is present simultaneously. Each expresses anti-microbial (maybe antiprotist) activity, so the anti-microbial (antiprotist) activity of a CFCM sample represents a cumulative activity. However, in an optimal test system, one would prefer to determine the anti-microbial activity of each of the single anti-microbial compounds one by one. 

Since the discovery that the activity (the “switch-on”/“switch-off” states) of those group genes (operons, biosynthetic gene complexes, and BGCs), which are responsible for the primary-secondary phenotypic phase shift in each known EPB strain, not only work under the control of the versatile regulator hexA [74,75], but are also coordinately regulated at a higher level (via *hfq*-gene controlled sRNS/HexA-mRNA base pairing). More precisely, the Hfq-dependent sRNA, ArcZ, directly base-pairs with the HexA-encoding mRNA [76,77,78]; it is possible to construct double-mutant strains from each EPB species (or isolate) following the genuine strategy called easyPACId (easy promoter-activated compound identification) approach [79]. Each biosynthesizes and releases only one single NR-AMP molecule into the culture media. This discovery may be a starting point of the renaissance of *Xenorhabdus* anti-microbial peptide research [76,77,78,79,80,81,82,83]. We managed to reconstruct the “easyPACId” *hfq* deletion mutant strains from EMA and EMC (Boros et al., in preparation) and publish here some information, including data about their anti-microbial potential. In this paper, we give an account of the details of constructing our *hfq*-del mutants from our EMA and EMC [29] strains and provide their phenotypic descriptions. Our “easyPACId” strains will be available for cooperation with fellow scientists worldwide. 

## 2. Materials and Methods

The plasmids [84,85,86,87,88,89,90,91,92] used in this study [77] are listed in Table A1 in Appendix B. Several different entomopathogenic bacteria (EPB) species were used as producers of anti-microbial compounds. Various clinical isolates of Gram-positive and Gram-negative bacteria served as test organisms, as did eukaryotic organisms (protozoa, fungi, and murine macrophages). Plasmids with a temperature-sensitive pSC101 replication system were produced in *E. coli* at 30 °C under antibiotic selection [84]. For cloning purposes, *E. coli* TG1 [85] was used as the host strain. For plasmid mobilization into *Xenorhabdus*, *E. coli* S17-1λ pir [86] was applied. *E. coli* strains were routinely grown in conventional Luria–Broth (LB) liquid media at 37 °C in the presence of the appropriate selective antibiotics. 

### 2.1. NR-AMP-Producing EPB Strains, Test Organisms, and Bioassays of CFCMs 

#### 2.1.1. The Anti-Microbial Activity of EPB CFCMs Was Assessed against Trypanosomatid Protozoa 

*Leishmania donovanii*, and *T. cruzi*, using both the native (i.e., centrifuged and sterile-filtered) and heat-sterilized (i.e., centrifuged, sterile-filtered, and autoclaved) CFCMs of liquid cultures of three species of anti-microbial-producing EPB strains, EMA, EMC, and EMK (*Xenorhabdus budapestensis* nov. DSM16342(T)) [28], *X. szentirmaii* nov. DSM16338(T)) [28], and *X. innexii* nov. DSM16337(T) [8,28,29] were bioassayed with promastigotes and epimastigotes of *Leishmania amazonensis* and *T. cruzi* (Brazil strain), respectively, as well as on clinical isolates of Gram-positive bacteria (methicillin-sensitive and -resistant *Staphylococcus aureus*) and Gram-negative bacteria (*Francisella novicida*, *E. coli*, *Salmonella typhimurium*, and *P. aeruginosa*), and with the yeast, *Candida albicans*. The J774 murine macrophage cells served as a control for measuring the general eukaryotic cytopathogenicity of the CFCMs [68] (see below and Table 1 for additional details). 

#### 2.1.2. The Anti-Protozoan Activity of EPB CFCMs 

EPB CFCMS were assessed against the protists. *H. meleagridis* of both the native and heat-sterilized (as previously described) CFCMs from liquid cultures of two anti-microbial-producing EPB strains (Pl.-Yellow and Pl.-RED). They were both grown in Medium 199 containing Earle’s salts, L-glutamine, 25 mM HEPES, L-amino acids, 15% heat-inactivated fetal bovine serum (FBS) and 0.22% rice starch for 65 h at 30 °C, representing the type color colony variants (yellow and red) of the EPB species *P. luminescens* TT01 [93] distinguishable on MacConkey [94] agar plates.

### 2.2. Bioassays

#### 2.2.1. Bioassays on *Leishmania*, *T. cruzi*, and Control Test Organisms

The bioassays were extended to clinical pathogenic protozoa using previously described routine methods [95]). The CFCM of EMA and EMC (stored at 4 °C for two years) were used for these experiments. Both native and autoclaved CFCM were tested in each bioassay. Samples of EMA-CFCM and EMC-CFCM were also fractionated using Centricon spin columns, as described by the manufacturer (Millipore, Sigma, Burlington, MA, USA), and fractions from <3 kDa and >3 kDa were assayed for activity. The bioassays were carried out in 96-hole tissue culture plates, in 200 µL volumes, containing 95 µL of the respective liquid media, 100 µL cell-free respective CFCM, and 5 µL of overnight LB cultures of the respective test organisms in duplicate. Briefly, colony count assays were used to measure activity against clinical isolates of bacteria and Candida. Overnight incubation in 25–100 microliters of a LB medium was followed by plating on a solid medium. MTT utilization assays were performed after overnight incubation of cultures of stationary phase pro- or epimastigotes, respectively, with CFCM (1:1 ratio) for 4 h for anti-leishmanial and -trypanosomal activity. Untreated controls were assayed in parallel with the LB medium alone. A reduction of CFUs (for bacteria or Candida) or MTT activity (for protozoa) of at least 80% was considered + for anti-microbial activity. Cytopathogenecity for mammalian cells was assessed using the macrophage cell line J774A.1 (TIB-67; ATCC, Manassas, VA, USA) in MTT assays and light microscopy. J774 cells [66,67] were grown in DMEM with high glucose (25 mM) and 1.5 g/L NaHCO3, 50,000 U/L penicillin 50 mg/L streptomycin, and 10% FCS (heat-inactivated). Cells were grown in bioreactor tubes in an incubator at 37 °C and 5% CO_2_. The J774 cells were passaged twice a week, similar to those described [68]. As for positive and negative controls, the CFCMs used in these experiments were only those that showed any kind of anti-microbial activity either against Gram-positive (*B. subtilis*) or Gram-negative (*E. coli* OP50) or *Candida* targets in previous experiments. The anti-microbial activities of each CFCM used in these experiments in Columbus (Ohio) had been determined in Wooster (Ohio, Klein Laboratory) right after they were obtained. Then, three years later, just before being used in *Leishmania* experiments in Columbus, that is, after a three-year store. No significant difference was detectable. In Columbus, each replicate was accompanied by the respective negative and positive control flasks, containing (i) only media, (ii) media + test organism without CFCM, and (iii) media plus CFCM only and without test organisms. The test organisms were naturally *Leishmania*, *T. cruzii*, *Candida*, MRSA, MSSA, *Pseudomonas*, *E. coli*, and J77, respectively.

#### 2.2.2. Bioassays on *H. meleagridis* Grown in Monoxenic Culture with *E. coli*

Each of the CFCMs mentioned above was tested on *Histomonas* grown in an in vitro monoxenic culture with *E. coli* DH5α cells. The monoxenic culture of *H. meleagridis* Turkey/Austria/2922-C6/04 with *E. coli* DH5α was grown in Medium 199 containing Earle’s salts, L-glutamine, 25 mM HEPES, L-amino acids, 15% heat-inactivated fetal bovine serum FBS, and 0.22% rice starch for 72 h at 40 °C, as described [52,59,60]. Consequently, the anti-*E.coli* activities also had to be determined. In the first experiment (see Appendix A), the anti-histomonal tests were cell-free filtrates from cultures of *P. luminescens* TT01 yellow and red variants, prepared as described above and tested on *E.coli* that were then plated, On the resective table (Table 2 in the Section 3 Results) we can see qualitative data (“sigle colonies”, “cell layers”).

As described in detail in the Appendix A, the liquid medium of the protozoan culture was removed and replaced either by different CFCMs (in different concentrations) (Experimental groups), or fresh media (Control groups). The *Histomonas* cell numbers were determined. This experimental was set up for trying to distinguish between the reasons for the death of the CFCM-treated *Histomonas* cells: whether they died by direct anti-protozoal effects of the EPB-released (“CFCM-born”) AMPs, or by the indirect consequences of the death of their mutualistic prokaryote partner *E. coli* DH5@ cells.

The unchanged protozoan cultures and protozoan cultures in which the culture medium was replaced by fresh Medium 199 supplemented with 15% FBS were used as controls. Each of the different CFCM analyses was performed in triplicate. The cultures were incubated at 40 °C and evaluated every 24 h by counting the protozoan cells. 

### 2.3. Construction and Bioassay of the Anti-Microbial Potential of hfq(del) Mutants of EMA and EMC

#### 2.3.1. Construction of *hfq*(del) Mutants of EMA and EMC (for Details, see Appendix B)

The Δhfq mutant EMC strain was generated according to the one-step gene inactivation method [87], which was adapted to *Photorhabdus* [88] and *Xenorhabdus.* The protocol was as follows: The appropriately modified plasmids (Table A1 in Appendix B, for plasmids) were introduced into *X. szentirmaii* (EMC) by electroporation. Electrocompetent cells were prepared for the introduction of plasmids as follows: 1 mL of *X. szentirmaii* (EMC) culture grown at 30 °C in a LB broth supplemented with ampicillin (Ap) (150 μg/mL) to OD600~1.1–1.2, centrifuged for 30 s at 9000 rpm at 2 °C. Then, resuspended in an ice-cold wash SMG buffer (0.5 M sorbitol, 0.5 M mannitol, 10% glycerol) and centrifuged again. The washing step was repeated two times; then, the cells were resuspended in a 60 μL SMG buffer. Approximately 1 μg of plasmid DNA purified by the Qiagen Plasmid Midi Kit according to the manufacturer’s protocol was added to 40 μL of electrocompetent cells. Electroporation was carried out using 2 mm gap electroporation cuvettes and a BTX Electro Cell Manipulator 600 with the setting 129 Ohm, 24 kV/cm. Transformants were selected on LB plates supplemented with the appropriate antibiotics (kanamycin (Km)—120 μg/mL, streptomycin (Sm)—100 μg/mL, gentamicin (Gm)—50 μg/mL, and Ap—150 μg/mL). For the *hfq* KO recombination, EMC cells containing plasmid pBZs7 were grown in a LB broth supplemented with the appropriate antibiotics to a late logarithmic (OD_600_~1.1–1.2) phase; then, the culture was 2× diluted with a fresh LB broth. L-arabinose was added in a final concentration of 1% and grown further in a shaker for 1.5 h at 30 °C. The electro-competent cells were prepared as described above. The KO PCR fragment was amplified using a pKD4 template plasmid and the primers hfq EMCdelfor and hfq EMCdelrev. The amplicon was precipitated in 96% ethanol, dried, and resuspended in 10 mM Tris pH 8.0. For electroporation, approximately 1 μg of a purified DNA fragment was used as described. After electroporation, cells were grown for 5 h at 30 °C, then spread on LB agar plates supplemented with 120 μg/mL Km and incubated at 30 °C for 2 days. The Δhfq::Km^R^ recombinant was tested by colony PCRs. For deletion of the Km^R^ cassette from the Δhfq::Km^R^ mutant, pBZs1 was electroporated. A Sm^R^Ap^R^Km^R^ transformant colony was grown overnight at 30 °C in a LB broth supplemented with Ap, spread on LB + Ap agar plates, and incubated for 24 h at 30 °C. Then, the colonies were individually tested for the loss of the Km^R^ gene and the pBZs1 plasmid on LB + Ap, LB + Km, and LB + Sm plates. The Km^S^ Δhfq strains were tested by colony PCR using the primer pair hfqseqfor–hfqseqrev. The Δhfq mutant EMA strain was generated according to [77]. The 769-bp upstream and the 617 bp downstream region of the hfq gene of EMA were amplified with primer pairs hfq_up-for-Bam-hfq_up-rev and hfq-EMA_down-for-hfq_down-rev-Sac, respectively. The two amplicons were assembled in the following PCR: The assembled amplicon was purified using the Illustra GFX PCR DNA and Gel Band Purification Kit (GE Healthcare), digested with BamHI and SacI, and ligated into the BamHI-SacI site of pBluescript II SK(+) (pBZS13). After sequencing, the BamHI-SacI fragment from pBZS13 was transferred into the BamHI-SacI site of the R6Kγ-based plasmid pBZs12, leading to the SacB-bearing mobilizable recombination vector pBZs17. The recombination vector pBZs17 was then mobilized into EMA from *E. coli* S17-1 λ pir. The ON culture of S17-1 λ pir/pBZs17 was 50× diluted with a fresh LB broth supplemented with Km and Sm and grown 3 h to OD600~0.5, while the EMA recipient was grown in LB + Ap ON to OD600~1.0. where 0.1 mL of donor and 1 mL of recipient cultures were mixed, centrifuged, and washed twice with 0.9% NaCl solution, then spread onto LB agar plates. After 4 h of incubation at 30 °C, the bacterial lawn was resuspended in a 3 mL 0.9% NaCl solution, and the cells were centrifuged and spread onto LB + KmAp agar plates. After four days of incubation at 30 °C, the Km-Gm-Ap-resistant recombinant colonies were tested by colony PCRs using primer pairs EMAhfqseqfor-trn5neo3 and pUCfor24-hfqseqrev. To gain Δhfq recombinants, the EMA::pBZs17 strain was grown for 18 h at 30 °C in LB + Ap broth, and 50, 100, and 200 μL of the culture was spread onto LB + Ap + 5% sucrose agar plates. After 48 h incubation at 30 °C, the individual SacR colonies were tested for KmS and GmS phenotypes; then, the appropriate colonies were tested by colony PCR using primers EMAhfqseqfor and hfqseqrev. To verify the *hfq* deletion, the amplicon was sequenced.

#### 2.3.2. Bioassay of the Anti-Microbial Activities of Wild-Type and *hfq*(del) Mutants of EMA and EMC

The bioassays were carried out in 96-well (8 rows and 12 columns) tissue culture plates, in 200 μL volumes, containing 95 µL Mueller–Hinton liquid media, 100 µL cell-free respective CFCM, and 5 µL of overnight LB cultures of the respective test organisms. Each test was carried out in duplicates in the following way: rows AB contained 100 µL wild-type CFCMs, 95 µL Mueller–Hinton (MH) media, and 5 µL of stationary phase-test organisms from 37 °C overnight cultures in each hole. Rows C and D contained 100 µL of the *hfq*-del mutant CFCMs of the same species (EMA or EMC), 95 µL Mueller–Hinton (MH) media, and 5 µL of stationary phase culture of the respective test organism (from 37 °C overnight cultures) in each hole. Row E: 100 µL wild-type CFCMs, 100 µL Mueller–Hinton liquid (MH) media. Row F: 100 µL of the *hfq*-del mutant CFCMs of the same species (EMA or EMC), and 100 µL of Mueller–Hinton (MH) media. Row G: 195 µL of 50% Mueller–Hinton (MH) media + 5 µL of stationary phase test organisms (from 37 °C overnight cultures) in each hole. Row H: Row G: 200 μL of 0% Mueller–Hinton (MH) media and 5 µL of stationary phase test organisms (from 37 °C overnight culture) in each hole.

### 2.4. Statistical Analysis

ANOVA was carried out using the respective propositions of SAS 9.4 (see Acknowledgment section). The significant differences (α = 0.05) between treatment means were assessed using the least significant difference (LSD).

## 3. Results

### 3.1. Results on Anti-Leishmanial Potential

Anti-leishmanial activities of the cell-free conditioned media (CFCM) of three species of the entomopathogenic–nematode–symbiont bacterium genus *Xenorhabdus*, *X. budapestensis* sp. nov (Lengyel) DSM16342 (T); *X. szentirmaii* sp. nov. (Lengyel) DSM16338 (T) and; *X. innexii* sp. nov., (Lengyel) DSM16336 (T), termed EMA, EMC, and EMK, respectively, were determined and compared with antibacterial and antifungal potential. Each of the three (EMA, EMC, and EMK) examined CFCM showed significant anti-leishmanial activities, but EMA and EMC exerted stronger anti-microbial potential in general than EMK. Data are summarized in Table 1.

Each of the three CFCMs strongly antagonized each of the tested clinical isolates of Gram-positive (methicillin-sensitive (MSSA), and -resistant (MRSA) *St. aureus*) and Gram-negative (*F. novicida*, *E. coli*, *S. typhimurium*, as well as *P. aeruginosa*) bacteria and a clinical isolate of *C. albicans* (Table 1).

As for the molecular sizes of the active natural fractions of the CFCM, interestingly, the anti-microbial activity against all microbes was retained in different size fractions of EMA (below 3 kDa) and EMC (above 3 kDa), indicating differences in the sizes of the active compounds in the different CFCMs. Autoclaved CFCM samples of EMA and EMC were as active as the native ones. As for EMK, only native CFCM samples were tested. The results demonstrated the thermostability of the active components in CFCM of EMA and EMC since autoclaving these samples did not affect the anti-microbial activity against any of the organisms. Interestingly, the anti-microbial activity of EMC, but not EMC, was susceptible to proteinase K (OK) inactivation. This suggests that the active compound (s) in EMC are proteinaceous, whereas in EMK, they are not. None of the CFCMs were toxic to J774 macrophages under the conditions in our study.

### 3.2. Antimicrobial Potential of EPB CFCM on the Prokaryotic and the Eukaryotic Partner of the H. meleagridis/E. coli Monoxenic Culture 

Each CFCM was tested on *H. meleagridis* cells, which (as discovered by [26]) can in vitro be cultured as monoxenic culture with *E. coli* DH5α cells, only. Consequently, the anti-Gram-negative bacterial activities were also determined.

#### Antibacterial Potential of EPB CFCM on the Prokararyotic Partner 

To see if the NR-AMP molecules were produced by strong AMP-producing EPB species, the anti-histomonal potential CFCMs from four different cultures were compared.

The *Photorhabdus* data are presented in Figure 1 (effects on *Histomonas*) and Table 2 (effects on *Escherichia coli* DH5α) cells. The results indicated that there is no difference between the anti-microbial/anti-*histomonal* potential of the colony color variants, as both exerted equally strong effects on both partners of the monoxenic culture. In the cultures supplemented with either the red or the yellow colony color variants of TT01, CFCM, the cell propagation ceased, and the *Histomonas* cells—earlier or later—died.

From Figure 1 and Table 2, we concluded that the tested CFCM of each of the two colony-color variants of the EPB *P. luminescence* exerted cytotoxic activity not only on the prokaryotic (*E. coli* DH5α) but also on the eukaryotic *H. meleagridis* Turkey/Austria/2922-C6/0 pathogen as well. Although thie experimental was set up was scheduled for trying to distinguish between the reasons for the death of the CFCM-treated *Histomonas* cells: whether they died by direct anti-protozoal effects of the EPB-released (”CFCM-born”) AMPs, or by the indirect consequences of the death of their mutualistic prokaryote partner *E. coli* DH5@ cells, these experimental results does not allow to take this kind of distintion.

The growth curve of the monoxenic culture from the non-supplemented (that is, only FKS-containing M199) control media (green) could be characterized as performing a perfect log phase after a moderate growth period. It starts at 24 h and reaches a peak between 48 and 72 h, after which it declines. The growth curve of the monoxenic culture grown in the “completely supplemented” M199 control media (blue) can be characterized by a log phase that starts right after the start of incubation (at 0 h), reaches a peak at 48 h, and then rapidly declines. Each of the exogenous EPB CFCM resulted in reduced *Histomonas* cells. The CFCM of the red variant acted immediately, while the yellow variants acted after 24 h. Even in the latter case, the size “growth peak” at 24 h was less than half of the 4 h value of the respective control cultures. As expected, the CFCM of each colony color variant almost totally killed all the cells of the *EMA* and monoxenic accompanying prokaryotic species *E. coli* DH5 alpha (Table 2).

### 3.3. Anti-Microbial Activities of the hfq(del) Mutants EMA and EMC

We have constructed hfq-del regulatory mutants similar to those of [79] (Bode et al., 2019) from both EMA and EMC strains. The relevant publication is in preparation and the method that we elaborated as a modified version of the original one is given in Appendix B. We then bioassayed the anti-microbial activities of the respective CFCMs on several targets. We intend to apply the recently discovered and ingenious method called easyPACId [k1] to identify antiprotozoal compounds in the future, similar to the work by the Turkish team of Prof. Selcuk Hazir and his associates at Menderes University in Ankara, Turkey [83]. This technique has been discovered and elaborated by Edna Bode and her associates in Frankfurt, Germany [79]. Briefly, ∆hfq mutants of two studied EPB bacterial species *X. budapestensis* and *X. szentirmaii* were constructed, and then, the CFCMs of the wt and hfq mutants were used to test different pathogen organisms. However, before obtaining that, we intend to forecast the limitations of the potential use of this approach. To learn how this influences the reproducibility of the method, we tested the anti-microbial potential of the hfq mutant of both EMA and EMC species, and the data are presented in Table 3, Table 4 and Table 5.

These data indicate that the extremely strong anti-Gram-positive potential of the EMA strain is likely, but probably not exclusively, under the control of the *hfq*-govern genetic regulation [76]. The anti-Gram-positive antibacterial potential of EMC is stronger than the anti-Gram-negative potential but, in comparison to that of the EMA, is weaker. Note that the EMC-resistant Gram-positive species are not the same as those resistant to the antimicrobials present in the CFCM of the *hfq*-del regulatory mutant of EMA. To see how the intraspecific variability modifies the picture, we compared the resistance/sensitivity patterns of different geographical isolations of the unambiguously EMA-sensitive Gram-positive bacterium species *Paeibacillus larvae* [96]. The *P. larvae* is a rapidly spreading deadly pathogen organism for honey bee larvae, so it is a significant pest of agricultural significance.

These data indicate that although the *P. larvae* species are uniformly and unambiguously sensitive to the anti-microbial(s) present in the CFCM of the EMA wild-type strain, there is a significant intraspecific variability concerning partly the resistance//sensitivity issue to EMC-produced antimicrobials, partly resistance/sensitivity issue to the antibiotic potential of EMA hfq-del regulatory mutants. These data are of key importance when drawing our conclusion concerning the usefulness and the limitation of the potential use of the easyPACId [79] approach in our future research. These data also confirm that the extremely strong anti-Gram-positive potential of the EMA strain likely, but probably not exclusively, is under the control of the *hfq*-govern genetic regulation [76]. As for the anti-histomonal studies, we sum them up in the following way.

In Figure 2, one can see the difference between the colonies of the wild type and the *hfq*-(the regulatory mutant of the *X. szentirmaii* DSM 16338 (EMC)). The spectacular physical difference is presented in the wild-type, left A, and the mutant, right B.

## 4. Discussion

When planning experiments aimed at applying AMPs against eukaryotic (for instance, anti-protozoal) pathogens, we have to take into consideration that the strategy of designing antiprotozoals to control eukaryotic parasites (pathogens) must be different from that of designing antimicrobials to control prokaryotic pathogens [97,98,99]. The targetable intracellular structures of a eukaryotic parasite (pathogen) biochemically might be similar to some intracellular structures of vital functions in the eukaryote host to be protected. 

As for our *H. meleagrididis* exeriments, we worked in a monoxenic system where prokaryotic and eukaryotics cells coexisted. We found that both the prokaryotic and the eukaryotic cells died when EPB CFCM were added in th right concentration.

The question is, however, that what was the primary reason for the death of the *H. meleagridis* cells. They might die (1) because their procaryote mutualistic partner died; (2) they might be killed by a direct antiprotist activity of any (or more) AMP molecules of the EPB CFCMs; or (3) the two killing effects cumulated. Unfortunately, these published experiments cannot give an unambiguous answer to that question. We hope for the answer from our future experiments when we intend to use CFCMs from double (easyPACId) mutants (with *hfq*-deletion and exchanged (re-activated) promoter of only one AMP-responsible operon. We will search for some which is harmless to prokaryotic partner but kill *H. meleagridis* cells selectively). It is true, that the peptides killed the mutualistic (*E. coli DH5@*) bacteria, which itself could be considered as an explanation of the death of the *H. meleagridis* cells. It is also true, that from our experiments no proof supports the idea that histomonads were killed directly. But we also have evidence that CFCM contains active anti-protist active components as well, (see the Leishmania donovanii data from this study or the published results of the Bode group and others), so the only question is what of these compounds is capable of (in lucky case, selectively) killing *H. meleagridis*. This is what we intend to learn from our future easyPACId mutant experiments.

This paper is about the anti-protist potential of anti-microbial compounds produced by entomopathogenic bacterium species (EMA, EMC, and EMK *Photorhabdus luminescens* TT01) [6,28,29,93], the obligate symbionts of certain entomopathogenic–nematodes (EPN). In laboratory in vitro conditions they release their products into their conditioned culture media (CFCM) in vitro. The chemical nature of these antimicrobials is oligo-peptide. However they are non-ribosomal anti-microbial peptides (NR-AMPs), which means that they are enzymatically synthesized through more than one step in their respective biosynthetic pathways. Those genes encoding for the enzymes forming the respective biosynthetic pathway of given NR-AMP are clustered in a single operon [100], also called biosynthetic gene complex (BGC) [101]. One of the natural roles of the prokaryote partner in the EPN/EPB symbiosis is to protect the given monoxenic symbiotic association in a given polyxenic milieu, producing a chemical toolkit (e.g., a particular set of NR-AMPs) providing competitiveness against the prokaryotic and eukaryotic competitors, including protists. The biosynthesis of the given sets of these effective antimicrobials is synchronized by a unique, genuine hierarchical genetic regulation mechanism called primary-secondary phase shift [71] with the *hfq* gene on the top [74,75,76,77,78]. In the laboratory liquid culture conditions (as well as in the active symbiotic stage), these compounds are produced abundantly, providing an inexhaustible “gold mine” for the anti-microbial searching scientist. The bioassays of the CFCMs are the tools to find the best sources and optimize the culture conditions. We found that those strains (EMA and EMC) which were discovered and further developed not only by our international team of a “laboratory without walls” in those famous laboratories of Maxime Gualtieri in France and Helge Bode in Germany, proving that these two strains are probably best found so far. In addition, other laboratories are probably really excellent sources. The data presented here on anti-leishmanial and anti-histomonal potentials seem to confirm this conclusion. The next step is to find a way to obtain mutants, not “genetically manipulated”, but mutant strains which produce only single NR-AMP molecules instead of a bunch of them. The method to do that was recently discovered in the Bode laboratory [79,80,81,82,83]. We demonstrate here that we have started this project as well by constructing and characterizing *hfq*-del mutants from both EPB species EMA and EMC.

Our larger scope is to benefit from the existence of NR-AMP amongst the metabolites of our EPB bacteria. In other terms, our goals are to beat target pathogens, including prokaryotic pathogenic and zoonic bacteria, and eukaryotic parasites and pathogens of different taxa, which are important from the aspects of importance for animal and human health.

We are keenly interested in exploring whether the anti-microbial activity of NR-AMP of EMA and EMC can be developed as chemotherapeutic agents to treat protozoa such as the intracellular parasite *Leishmania* [37] and the extracellular avian pathogen *H. gallinarum* [97]. Both of these flagellated protozoa have been demonstrated to have unique and efficient resistance mechanisms to other anti-parasitic agents and require the development of additional effective agents. *Leishmania* spp. is particularly problematic in light of its sophisticated resistance mechanisms, but it is also an excellent model for analyzing the mechanisms of drug resistance in eukaryotes [98,99]. Our research conception is that we think that some *Xenorhabdus* CFCM molecules (i.e., phenazine, fabclavine, or other molecules) may be selectively capable of dysregulating eukaryotic cell functions and stimulating apoptosis in protozoa without affecting the host [100]. From an application point of view, the development of natural compounds as anti-protozoal requires several aspects to be taken into consideration: (A) the anti-protozoal potential; (B) the durability, thermotolerance, and bioavailability; and (C) cytotoxicity, and unwanted side effects, especially if target organisms are eukaryotic pathogens. We have demonstrated that *Xenorhabdus* CFCM has broad-spectrum anti-protozoal activity against several human microbial pathogens. The key issue is identifying the specific active molecules in a complex mixture of potential anti-microbial molecules. The CFCMs of all three *Xenorhabdus* species were all active against *Leishmania* promastigotes in our experiments, but interestingly, none were active against *Trypanosoma cruzi* epimastigotes. Other researchers [83] found that EMA and EMK CFCMs were equally efficient against both *Leishmania* and *Trypanosoma*. In that study, they found that *X. budapestensis*, *X. cabanillasii*, *X. hominickii*, *X. indica*, *X. innexii*, and *X. stockiae* supernatants caused 100% mortality at the highest tested concentration (10%) against the promastigote form of *L. tropica*. No differences occurred between this treatment group and the positive control (*p* > 0.05) using N-methyl meglumine. There are several possible explanations for the differences between our findings and those of the Turkish team. For instance, the two studies used different strains of *T. cruzi*. In addition, there are differences in the experimental methods used between the two studies. In the case of the differential susceptibility of *L. amazoennsis* and *T. cruzi* (Brazil strain), it is likely due to differences in the surface compositions between the two parasites that lead to differential disruption of the cell membrane and/or intracellular penetration to act on specific subcellular targets. We focus on EMA and EMC species. We compared the anti-microbial potential of each of the used *hfq*-del mutants before using them as the source of double mutants producing single NR-AMPs. Our data in Table 3, Table 4 and Table 5 show that the anti-microbial activities of the *hfq*-del mutants can have enhanced anti-microbial activity not present in the wild-type stains. This may not be quite surprising considering that approximately 7.5% of the total genes in *Xenorhabdus* bacteria are dedicated to secondary metabolite biosynthesis, and probably most of these products are NR-AMPs. With an average 2.4 Mb-genome size of *Xenorhabdus* and a 1 kb locus size with about 10 ORFs per operon, there are 2400 genes and 240 respective operons (BGS) per genome. It is comparable with that of *Staphylococcus*, which has about 225 operons. According to the data accumulated, only about a dozen of biosynthetic anti-microbials have been discovered. In our lab, we intend to work with about 12 operons, responsible for the biosynthesis of 12 NR-AMPs in EMC, and operons which are responsible for 3 NR_AMPs in EMA under *hfq* regulations.

However, we cannot rule out that there are several other operons encoding for the synthesis of additional anti-microbial compounds, which are not under hfq control. In the future, we intend to establish and analyze RNA-seq from our wild-type stains and their *hfq* mutants. We hypothesize that the situation is somewhat simpler since the hfq likely regulates a well-defined group of operons providing the genetic machinery for the so-called “primary-secondary phase shift” [72] (or “phenotypic phase variation”, the term of Professor S. A. Forst), which describes that only a relatively small set of natural products (NP) are anti-microbial synthetic genes. Toxicity studies using J774 macrophages [66,67] have indicated that only the CFCM of EMA was able to cause significant mortality of these cells. In contrast, the CFCMs of EMC or EMK had no effect. This experiment confirmed our previous unpublished observations that EMA exerts stronger cytotoxicity on a range of eukaryotic cells, which may cause limitations concerning future medical applications. However, this selective cytotoxicity may provide an option to select for and find specific CFCMs against different eukaryotic parasites of different animal and plant hosts. Interestingly, our studies with J774 cells and the CFCMs of EMA, EMC, and EMK did not show toxicity. These differences may be related to different experimental methods or differences in the bacterial and cell types used.

Overall, in light of the availability of few agents for the treatment of leishmaniasis and the increasing incidence of development resistance, there is a demand for new agents [100,101,102,103]. Moving forward, drug discovery will be based on complete genome sequencing of multiple strains, utilization of CRISPR/Cas9 technology for gene inactivation, in vivo, and bioluminescence-based imaging together with high-throughput analysis of compounds [44]. As for the results in the control (bacterium, fungi, and other strains), previous studies have identified several anti-microbial molecules in the CFCM of *Xenorhabdus*, including fabclavine [25,26]. This strange metabolite polypeptide/polyketide hybrid molecule is synthesized via non-ribosomal peptide synthesis. Many different-sized derivatives of fabclavines can be produced by *X. szentirmaii* [80] and can vary between entomopathogenic–nematode–symbiont (*Xenorhabdus*, *Photorhabdus*) bacterium species [80] (Wenski et al., 2019). These have a wide-ranging anti-microbial activity against various microbes [29]. Several other anti-microbial molecules are known to be produced by different EPB species and released into their CFCMs. Xenofuranone A and B [104], cabanillasin [105], PAX peptides [106], odilorhabdins [107], cyclic depsipeptides (xenematides, F and G), anti-oomycete peptides [108], novel anti-microbial peptides [109], xenortide [110], xenortides, rhabdopeptided/xenortide-like peptides [111], other rhabdopeptides [112], and szentiamid [113,114] have been discovered in the last ten years in different *Xenorhabdus* species.The proper approach is the easyPACId method, based on the latest information on the highest organization level of gene regulation in bacteria [115,116,117,118,119,120]. Whether any of them can be considered a potential anti-histomonal drug candidate will be learned if we had a chance to screen both the single mutant *hfq-*del stains of EMA and EMC.

## 5. Conclusions

Conclusions could be drawn from this study if we see the aims and the results along with the following aspects:Perspectives of EPN/EPB symbiotic associations as abundant sources of AMPs, and we feel that the present publication provides some new idea into this directionThe comparative aspects of biosynthetic (non-ribosomal templated) and primary gene-product AMPs from drug potential aspects; and this publication drew the attention of the reade tosome complementary aspects.The strategic aspects of controlling selected eukaryotic pathogens; we concluded the NR-AMPS have more option of becoming potential drugs to handle this issue.The scientific and industrial aspects of the recently discovered hierarchical regulatory systems following signal integration provide great metabolic versatility that results in excellent adaptability and metabolic optimization in antibiotic-producing bacteria [99], like EPBs.

We would like to emphasize the actual importance of this research when concluding that the delineation of specific anti-microbial products in CFCM can now be achieved using the “easyPACId” approach (easy promoter-activated compound identification), which takes advantage of the alteration of regulation of coordinate gene clusters whose products are responsible for their biosynthesis [79]. We are convinced that this discovery may revolutionize the *Xenorhabdus*/*Photorhabdus* research, which in the age of multidrug resistance seems important. The point is that the operation of the different biosynthetic gene clusters BGCs (that is, operons) is synchronous, and their regulation is coordinated [76,77] by the gene encoding the RNA chaperone, Hfq [79]. The experimental system is based on the option of manipulation of the regulatory gene *hfq* [100,101,102,103,104] of the respective species. The regulatory mutant strain does not produce any biosynthetic AMP, but the respective individual promoters upstream of the respective biosynthetic gene cluster can be changed to inducible promotors, and this recombination will produce only a single respective AMP molecule. This provides an option of obtaining bacteria that produce only one single AMP molecule. The NR-AMP molecules have similar advantages to traditional antibiotics as the ribosomal templated AMPs, that is, the different pools of resistance genes with different motility patterns and collateral sensitivity, as well as molecular versatility. The advantage of the natural abundance in AMPs provided by the coevolutionary pattern of EPN/EPB symbiotic association may provide ammunition usable against MDR pathogens for a long time. This research line is further facilitated by the completed genome sequencing of several *Xenorhabdus* species, including *X. szentirmaii* [121] and *X. budapestensis* [122], making possible the identification of the promoter regions of each biosynthetic gene cluster (BGC, operon). Despite demonstrating the efficacy of vaccination in an experimental setting [55,58,123,124,125] no vaccine is yet commercially available. Therefore, and in addition to various other attempts, alternative chemotherapeutic solutions ares justified and the application potential of AMPs [126] seems a realistic idea.

Creating bacterial mutants that synthesize individual anti-microbial products into CFCM can facilitate their identification, purification, and bioassays of anti-microbial activity [82] and finally their potential application.

## Figures and Tables

**Figure 1 antibiotics-12-01462-f001:**
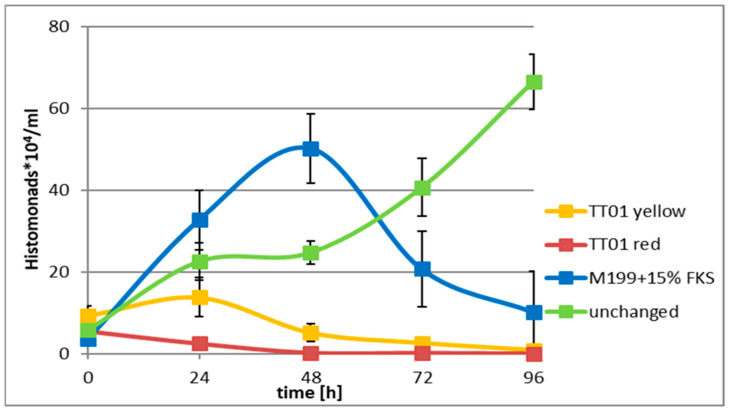
Monoxenic culture: *Histomonas*
*meleagridis* (Turkey/Austria/2922-C6/04) cells were co-cultured with *E. coli* DH5α cells not-supplemented M199 tissue culture media, which contained FKS (green, or n supplemented M199 tissue culture media (blue). The cells were viable and propagated up to 72h in (**a**) (green) or at least 48 h in (**b**) (blue) at 40 °C in the CFCM-free cultures. In those cultures, however, which were supplemented with the sterile filtered CFCM (either of the red or the yellow) TT01 colony-color variants, the cell propagation quickly ceased, both in the supplemented M199 media (red, earlier) and in the non-supplemented (but FKS-containing) M199 media, a little later, and then, the *Histomonas* cells died (data are given in the Appendix A).

**Figure 2 antibiotics-12-01462-f002:**
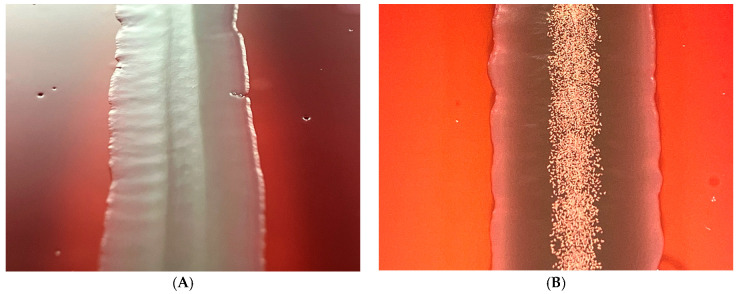
A colony of the *hfq*-del regulatory mutant (**left**, **A**), and that of the the wild-type (**right**, **B**) and of *Xenorhabdus szentirmaii* DSM16339 (EMC). The spectacular iodine crystals (the end product of the phenazine biosynthesis) are missing, indicating that the phenazine-operon encoded biosynthesis pathway is switched off in the *hfq*-deleted regulatory mutant.

**Table 1 antibiotics-12-01462-t001:** Anti-microbial activities of 5-day mid-stationary-phase cell-free conditioned culture media (CFCM) of Xenorhabdus species *.

CFCM	EMA					EMC					EMK
Pre-Treatment:	None	Heat	PK	Centricon		None	Heat	PK	Centricon		None
				<3 kDa	>3 kDa				<3 kDa	>3 kDa	
Bacteria-Gram +											
*Staphylococcus aureus (MRSA)*	+	+	−	+	−	+	+	+	−	+	+
*Staphylococcus aureus (MSSA)*	+	+	−	+	−	+	+	+	−	+	+
**Bacteria-Gram −**											
*Francisella novicida*	+	+	−	+	−	+	+	+	−	+	+
*Salmonella typhimurium*	+	+	−	+	−	+	+	+	−	+	+
*Escherichia coli*	+	+	−	+	−	+	+	+	−	+	+
*Pseudomonas aeruginosa*	+	+	−	+	−	+	+	+	−	+	+
**Protozoa**											
*Leishmania amazonensis*	+	+	−	+	−	+	+	+	−	+	+
*Trypanosoma cruzi*	−	−	−	−	−	−	−	−	−	−	−
**Fungi**											
*Candida albicans*	+	+	−	+	−	+	+	+	−	+	+
**Toxicity**											
J774 macrophages	−	−	−	−	−	−	−	−	−	−	−

* Notes: CFCM = 5-day mid-stationary-phase cell-free conditioned culture media of *Xenorhabdus* species; EMA = X budapestensis nov. DSM16342T; EMC = *X. szentirmaii* nov. DSM16338T; EMK = *X. innexii* nov. DSM16336T; Heat = autoclaved in standard conditions (15 min, 121 °C, 1 Atm) and then tested; PK = indicated CFCM was pre-incubated with proteinase K (1 mg/mL for 30 min at 37 °C) prior to testing for anti-microbial activity; Centricon = molecular size fractionation of indicated CFCM in centricon spin columns prior to testing anti-microbial activities; Toxicty = host toxicity, tested on determined microscopically and also determining MTT metabolism of J774-line murine macrophages (serving as checking for unspecific general cytotoxicity), exposed to the respective CM at 37 °C for 72 h; MS and MR: methicillin-sensitive and -resistant *Staphylococcus aureus*, respectively. Activity assays were performed on cultured cells of the respective bacterial and fungal microorganisms incubated in 25–100 microliter of a Luria–Bertani liquid medium overnight. Each was then plated on the appropriate medium to CFU reduction assays (concerning bacterial or fungal pathogens) or using MTT utilization for protozoa. +, anti-microbial activity with greater than 80% reduction in growth after CFCM incubation compared to untreated control cells; −, no anti-microbial activity observed.

**Table 2 antibiotics-12-01462-t002:** Effect of *Photorhabdus* CFCMs on the prokaryote partner of *Histomonas*
*meleagidis* in monoxenic culture.

			*E. coli* DH5 Alpha			
Sample	Hours	0	24	48	72	96
1	TT01 Yellow		Single colonies	Single colonies	Single colonies	Single colonies
2	TT01 Yellow		Single colonies	Single colonies	Single colonies	Single colonies
3	TT01 Yellow		Single colonies	Single colonies	Single colonies	Single colonies
4	TT01 Red		Single colonies	Single colonies	Single colonies	Single colonies
5	TT01 Red		Single colonies	Single colonies	Single colonies	Single colonies
6	TT01 Red		Single colonies	Single colonies	Single colonies	Single colonies
7	M199 + 15% MKS		Cell layer	Cell layer	Cell layer	Cell layer
8	M199 + 15% MKS		Cell layer	Cell layer	Cell layer	Cell layer
9	M199 + 15% MKS		Cell layer	Cell layer	Cell layer	Cell layer
10	Unchanged		Cell layer	Cell layer	Cell layer	Cell layer
11	Unchanged		Cell layer	Cell layer	Cell layer	Cell layer
12	Unchanged		Cell layer	Cell layer	Cell layer	Cell layer

100 μL samples were taken from *H. meleagridis*/*Esherichia coli* monoxenic culture (grown in M199 media) and plated on MacConkey agar afterward, where only the *E. coli* cells can grow, while the *H. meleagridis* cells cannot.

**Table 3 antibiotics-12-01462-t003:** Anti-Gram-negative activities of CFCMs of wild-type n *hfq* (del) regulatory mutant cells of *Xenorhabdus budapestensis* DSM16342 and *X. szentirmaii* DSM 16338 (EMC).

TEST ORGANIS?	EMA (WT)	EMA (hfq-del)	EMC (WT)	EMC (hfq-del)
GRAM-NEGATIVE TARGETS				
*E. coli*	−	**−**	−	+
*Klebsiella pneumoniae*	−	**−**	~50%	+
*Salmonella enterica*	−	+	+	+
*Pasteurella multocida*	−	**−**	+	+
*Mannheimia haemolytica*	−	+	+	+
*Ps. aeruginosa* VB (phytopathogenic)	−	~50%	+	+
*Actinobacillus equuli*	−	+	+	+
*Enterobacter cloacae*	−	+/−	+	+
*Serratia marcescens*	−	+	+	+
*Bordetella bronchiseptica*	−	−	+	+

Each of the tested Gram-negative bacteria of veterinary significance was completely sensitive to one or more anti-microbial compounds in the CFCM of the wild-type *X. budapestensis* DSM16342 (EMA) strain. The majority of them (except for *E.coli*, *Salmonella*, and *Bordetella*) were resistant to the anti-microbial active compound(s) present in the CFCM of the *hfq* (del) regulatory mutant of the same species (EMA). All but *E. coli* and *Klebsiella* were sensitive to the anti-microbial ingredient present in the CFCM of the wild-type *X. szentirmaii* DSM 16338 (EMC), but each of them was fully resistant to the anti-microbial active compound(s) present in the CFCM of the *hfq* (del) regulatory mutant. Abbreviations: −: The test organism grew in the given CFCM, so the test organism is considered resistant to the antimicrobials in the CFCM present; +: The test organism did not succeed in the given CFCM, so the test organism is considered sensitive to the antimicrobials in the CFCM present; +/−: the test organism seemingly but hardly grew at least in one of the replicates, so the test organism is considered as somewhat resistant to the antimicrobials in the CFCM present. These data indicate that the strong anti-Gram-negative potential of the EMA strain likely, but probably not exclusively, is under the control of the *hfq*-govern genetic regulation [76]. The anti-Gram-positive potential of EMC is limited in comparison to the EMA. It is similarly true concerning the anti-Gram-positive antibacterial potential; see Table 3.

**Table 4 antibiotics-12-01462-t004:** Anti-Gram-positive activities of CFCMs of wild-type n *hfq* (del) regulatory mutant cells of *Xenorhabdus budapestensis* DSM16342 and *X. szentirmaii* DSM 16338 (EMC).

TEST ORGANISMS	EMA (WT)	EMA (*hfq*-del)	EMC (WT)	EMC (*hfq-del*)
GRAM-POSITIVE				
*Bacillus cereus*	−	−	−	NT
*Staphylococcus pseudintermedius*	−	−	+	NT
*Staphylococcus aureus* (MRSA)	−	−	+	NT
*Staphylococcus hyicus*	−		−	NT
*Streptococcus suis*	−	+	−	NT
*Streptococcus agalactiae*	−	−	−	NT
*Listeria monocytogenes*	−	+	−	NT
*Erysipelothrix rhusiopathiae*	−	−	−	NT
*Corynebacterium pseudotuberculosis*	−	−	−	NT
*Rhodococcus equi*	−	−	−	NT

It can be seen that each of the tested (pathogenic and/or zoonic) Gram-negative bacteria was completely sensitive to one or more anti-microbial compounds in the CFCM of the wild-type *Xenorhabdus budapestensis* DSM16342 (EMA) strain. The majority of them (except for *E.coli*, *Salmonella*, and *Bordetella*) were resistant to the anti-microbial active compound(s) present in the CFCM of the *hfq* (del) regulatory mutant of the same species (EMA). All but *E. coli* and *Klebsiella* were sensitive to the anti-microbial ingredient present in the CFCM of the wild-type *X. szentirmaii* DSM 16338 (EMC), but each of them was fully resistant to the anti-microbial active compound(s) present in the CFCM of the *hfq* (del) regulatory mutant. Abbreviations: −: The test organism grew in the given CFCM, so the test organism is considered resistant to the antimicrobials in the CFCM present; +: The test organism did not succeed in the given CFCM, so the test organism is considered sensitive to the antimicrobials in the CFCM present; +/−: The test organism seemingly but hardly grew at least one in one of the replicates, so the test organism is considered as somewhat resistant to the antimicrobials in the CFCM present.

**Table 5 antibiotics-12-01462-t005:** Intraspecific variability within the Gram-positive *Paenibacillus larvae* to anti-microbial compounds of hfq deleted mutants.

TEST ORGANISMS	EMA (WT)	EMA (*hfq*-del)	EMC (WT)	EMC (*hfq-del*)
GRAM-POSITIVE				
*Paenibaillus larvae* isolates				
ML-234	−	+	+	+
ML-236	−	+	+	+
ML-237	−	+	+	+
ML-238	−	−	+	+
ML-239	−	+	+	+
ML-248	−	−	−	+
ML-250	−	−	−	+
ML-252	−	−	−	+
ML-263	−	−	−	+

Abbreviations: +: The test bacterium grows, −: The test bacterium does not grow. Each of the tested geographical isolates of the Gram-positive (deadly pathogen for honeybee larvae) *Paeibacillus larvae* [96] bacterium species seems to be completely sensitive to (one or more) anti-microbial compounds present in the CFCM of wild-type *Xenorhabdus budapestensis* DSM16342—mans (EMA). Six of the nine isolates, however, also seem to be sensitive to the anti-microbial active compound(s) present in the CFCM of the *hfq* (del) regulatory mutant of the same species (EMA). About half of them (mainly those which are resistant to the antimicrobials of the *hfq* (del) regulatory EMA mutant) were also resistant to the wild-type EMC CFCM, but each of them is resistant to the antimicrobials present in the CFCM of the *hfq*/del, a regulatory mutant of EMC.

## Data Availability

All of the data are available from the corresponding author. All data generated from this study are included in this article, and some information in the Appendix A is strongly inseparable from information published in our previous paper [6].

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
