# Peer review of "Antimicrobial Peptides (AMP) in the Cell-Free Culture Media of Xenorhabdus budapestensis and X. szentirmaii Exert Anti-Protist Activity against Eukaryotic Vertebrate Pathogens including Histomonas meleagridis and Leishmania donovani Species"

_antibiotics, 2023, doi:10.3390/antibiotics12091462_

Round 1
Reviewer 1 Report
Dear authors ,
It is a really great work.
As a great work it should be in the appropriate level in the manuscript. The manuscript looks as a draft... before the final submission version...
Please homogenize the fonts, the paragraph style, etc in the whole manuscript. i.e. page 8. It is a page with several font, size, spaces, etc...
Please correct the several typos...
The same occurs in references... please correct them...
The experiments are very well presented. However, a negative or positive control is missing, it is useful especially when you compare antimicrobial or antiviral activity... Please explain
You have to choose carefully which results are going to be presented in the final manuscript and which are going to be presented in the supplementary file... please consider them
Reviewer 2 Report
Reviewer comments- antibiotics-2561092
This manuscript describes “Antimicrobial Peptides for Combating Eukaryotic Pathogens: Anti-Histomonal, and Anti-Leishmanial Potential of Antimicrobial Peptides Released by Entomopathogenic Nematode Symbiont Bacteria.” This is an interesting study on antimicrobial peptides for its medicinal potential as Anti-Histomonal, and Anti-Leishmanial agents. However, there are some major and minor issues in the current manuscript. After addressing following concerns, this article can be considered for publication.
Major and Minor concerns:
· Title is too long. Please revise it. It can be written such as ; “Entomopathogenic Nematode Derived Antimicrobial Peptides As Anti-Histomonal, and Anti-Leishmanial Agents” Or something like this.
· In introduction, as this article related to peptides (antimicrobial peptides) and its utility as biomedical applications, so it would be better if authors can add some discussion about peptide drug discovery. As there is already number of peptide-based drugs (>100 drugs in clinic) already in clinic so some discussion would increase readers interest to this peptide work. Please see reference: https://doi.org/10.1016/j.drudis.2022.103464 (Drug Discovery Today (2022): 103464.) and https://www.sciencedirect.com/science/article/pii/S1359644614003997 (Drug discovery today 20, no. 1 (2015): 122-128.); and these can be cited.
· Table 1, it would be better if author can add name of strain itself in table for higher visibility. Such as MS FN ST EC are not appropriate.
· Figure 1, error bars are missing.
· Conclusions need serious improvements as this is not conclusion. It needs to be systematically explained this article.
· Author Contributions need to be written with more clarity as it is not suitable correctly. Please mentioned name of studies performed by respective authors and their role.
· All reference should in uniform pattern as multiple patterns used.
NA
Reviewer 3 Report
The introduction is a bit long and could be shortened. Some of the sentences are unnecessary and could be removed.
On line 154 to 155, you mentioned that "since their introduction into the market in the middle of the previous century, further research nearly ceased as outbreaks of histomoniasis occurred only very rarely." Why are you still studying this histomaniasis? Or Why did you mention this?
What are the reasons that these drugs were banned? (e.g. line 157-158, line 217-219)
Line 250-252, what about the cell lines in the Trypanosoma study?
What are the sequences of the primers used?
In table 1, there are two instances of the abbreviation "MS." One of these instances should be "MR." Please correct this error.
Table 3&4, can you include data such as percent of antibacterial potential instead of +/-. This would be helpful to the reader in understanding the efficacy of the peptides against the different bacteria.
Line 57 needs references
There are several run-on sentences in the article, for example, 243-245, 289-291, 640-644 et al. The authors should break these sentences up into two separate sentences
Also, please correct the grammar, abbreviations, font, and other things like that.
Round 2
Reviewer 1 Report
The authors responded in most of my comments.
My concern is for the presentations of the tables. To my opinion it is better to present them in landscape form instead of portrait if there are too long.